# The impact of induced stress on reactive and proactive control in depression

Akihiro Masuyama

Department of Psychology, Aichi University of Education, Kariya, Aichi, Japan

## ABSTRACT

**Background:** Depression, a widespread mental health issue, is often marked by impaired cognitive control, particularly in managing proactive and reactive processes. The Dual Mechanisms of Control (DMC) framework differentiates between these two modes of cognitive control: proactive control involves sustained goal maintenance, while reactive control is more stimulus-driven and transient. Stress, known to exacerbate cognitive dysfunction in depression, may influence the balance between these control processes, though the specific effects remain poorly understood. This study aimed to investigate how acute stress influences proactive and reactive control in individuals with depressive symptoms.

**Methods:** A total of 142 participants were divided into high-stress and control conditions and further categorized based on their depression levels, measured using the Beck Depression Inventory-II (BDI-II). Cognitive control was assessed using the AX-Continuous Performance Task (AX-CPT), both before and after exposure to a stress-inducing anagram task, which was designed to differentiate between high-stress and low-stress conditions.

**Results:** Participants exposed to the high-stress condition reported significantly greater stress and fatigue levels compared to the control group, validating the stress manipulation. Although the balance between reactive and proactive control, as measured by the Proactive Behavioral Index (PBI), did not show significant changes, depressive individuals in the high-stress condition exhibited a significant decline in their ability to retain contextual information, as indicated by a reduction in the $d'$-context index. This suggests that depressive individuals may be more prone to stress-induced difficulties in proactive control.

**Discussion:** These findings highlight the selective impact of stress on proactive cognitive control in individuals with depressive symptoms, shedding light on a potential cognitive vulnerability in depression. While the balance between reactive and proactive control remained stable, the impaired retention of contextual information post-stress points to a specific deficit in proactive control. This could have implications for targeted cognitive interventions, such as cognitive control training, aimed at enhancing resilience against stress in depressive populations. Future research should explore the long-term effects of stress on cognitive control, particularly in clinically diagnosed individuals.

Corresponding author
Akihiro Masuyama,
ak.masuyama@gmail.com

How to cite this article Masuyama A. 2025. The impact of induced stress on reactive and proactive control in depression. **PeerJ 13:e18821**
**DOI 10.7717/peerj.18821**

## INTRODUCTION

Depression, characterized by a persistent low mood and negatively biased cognitive patterns, remains a widespread mental health concern globally. According to the *World Health Organization (2023)*, an estimated 3.8% of the global population suffers from depression. Along with its high prevalence, the recurrence rate of depressive episodes is notably high (*Buckman et al., 2018*). These factors underscore the need for a more comprehensive understanding of depression's cognitive underpinnings. One key aspect is the cognitive impairments associated with depression, particularly those involving negatively biased thoughts and impaired cognitive control (*Gotlib & Joormann, 2010*).

Cognitive control refers to the ability to maintain and update task goals and behavior in response to changing demands, allowing for adaptive cognitive and behavioral responses (*Miller & Cohen, 2001*). It encompasses a range of functions and has been conceptualized through various models. For instance, *Botvinick et al. (2001)* proposed a conflict-monitoring model, which emphasizes the role of detecting and resolving conflicts in cognitive processing. On the other hand, *Miyake et al. (2000)* outlined a three-component framework of executive functions, including shifting, updating, and inhibition, which highlights the diversity of processes involved in cognitive control. Among these, the Dual Mechanisms of Control (DMC) framework (*Braver, Gray & Burgess, 2007*) distinguishes between two modes of cognitive control: reactive control, which operates in a bottom-up fashion by responding to stimuli as they occur, and proactive control, a top-down process that involves maintaining goal-relevant information over time (*Braver, 2012*; *Braver, Gray & Burgess, 2007*; *Braver et al., 2021*). The DMC framework has been influential in understanding cognitive control in various contexts, including aging (*Kopp et al., 2014*), schizophrenia (*Kwashie et al., 2023*), and attention deficit hyperactivity disorder (*Cai et al., 2023*).

Several studies have demonstrated that depressive symptoms are closely linked with deficits in cognitive control (*Joormann & Tanovic, 2015*; *Snyder, 2013*; *Yang, Yang & Isen, 2013*). For instance, *Snyder (2013)* conducted a meta-analysis highlighting impairments in working memory and inhibitory control in individuals with depression, while *Yang, Yang & Isen (2013)* emphasized the role of these deficits in perpetuating depressive rumination. These cognitive control deficits are thought to contribute to various depression-related phenomena, such as attention and memory biases, rumination, and automatic negative thoughts (*Gotlib & Joormann, 2010*). *Gotlib & Joormann (2010)* further elaborate on how these biases and negative thought patterns emerge from impairments in cognitive flexibility and goal-directed processing. The studies within the DMC framework have shown that individuals with depression tend to rely more on reactive control and less on proactive control during tasks that require both (*Msetfi et al., 2009*). Additionally, reactive control is further impaired when individuals with depression are exposed to negative emotional stimuli (*Masuyama et al., 2018*). Mood induction procedures have also demonstrated that negative mood exacerbates proactive control deficits in depressive individuals, suggesting an imbalance between the two control processes

(*Masuyama & Mochizuki, 2020*). While these findings are valuable, many questions remain, particularly regarding how stress affects cognitive control processes in depression.

Stress is a well-documented factor influencing cognitive function, particularly in the context of depression. Stress has been shown to impair cognitive control across various domains (*Starcke, Agorku & Brand, 2017*; *Vargas et al., 2020*; *Vrshek-Schallhorn, Velkoff & Zinbarg, 2019*). For instance, studies have found that cognitive control can predict how individuals respond to stressful life events (*Gabrys et al., 2018*). In experimental settings, individuals with depression tend to perform worse on cognitive control tasks after being exposed to stress (*Quinn & Joormann, 2014*). Moreover, cognitive control has been found to predict rumination and stress responses, even after a short delay (*Zareian, Wilson & LeMoult, 2021*), and cognitive control training has been shown to enhance resilience to stress (*Hoorelbeke et al., 2015*). These results suggest that stress may have a profound impact on cognitive control, particularly in individuals with depression. While several studies have investigated the effects of stress on cognitive control, the specific impact of stress on the balance between reactive and proactive control within the DMC framework remains unclear. For example, a study by *Husa, Buchanan & Kirchhoff (2022)* found that physiological changes induced by stress led to a predominance of proactive control. However, this study did not focus on individuals with depression, and the effects of stress on reactive and proactive control in depressive populations are still not fully understood.

Therefore, the purpose of this study was to investigate the impact of experimentally induced stress on reactive and proactive control in individuals with depressive symptoms. Specifically, I aimed to determine whether the balance between these two modes of control, as described by the DMC framework, changes after exposure to an experimental stressor. The experimental stressor used was an anagram task, a method widely adopted in previous studies (*Yang et al., 2018*). The anagram task was chosen for its versatility and well-documented efficacy in inducing moderate stress levels in a controlled manner. Unlike other stress-induction paradigms, such as the Trier Social Stress Test (TSST), which may require extensive setup and interpersonal interaction, the anagram task is simple to administer in online settings, ensuring consistency across participants. Furthermore, this task allows for precise manipulation of stress intensity through the difficulty of the anagrams, enabling a more tailored stress induction appropriate for the study's aims. Previous studies have shown that tasks like the anagram task effectively evoke stress-related cognitive and emotional responses (*Starcke, Agorku & Brand, 2017*). By examining the effects of stress on cognitive control in depression, this study seeks to contribute to a deeper understanding of the cognitive vulnerabilities associated with depression and may inform new treatment strategies targeting cognitive processes, such as cognitive control training and attentional bias training (*Hoorelbeke et al., 2015*; *Maciejewski et al., 2020*).

## MATERIALS AND METHODS

Portions of this text were previously published as part of a preprint (https://doi.org/10.31234/osf.io/dgph3).

## Participants

A total of 142 individuals (55 female, 84 male, three unspecified) participated in this study (mean age = 40.46, SD = 7.99). Participants were recruited through a cloud-sourcing service (Crowdworks: https://crowdworks.jp). Eligibility criteria included being between the ages of 18 and 65, having normal or corrected-to-normal vision, access to a stable internet connection, and a desktop or laptop computer, as well as a distraction-free environment during the tasks. All study procedures were conducted using the online experiment platform Pavlovia (https://pavlovia.org), where participants completed the tasks on desktop or laptop computers with sufficient internet speed. Participants received an explanation of the study online, and informed consent was obtained electronically. In addition to providing informed consent for participation in the study, participants were required to ensure the following conditions in advance: 1) access to a stable and adequate Internet environment, 2) use of a designated web browser (Google Chrome or Firefox), and 3) participation in the experiment in an environment free from distractions. Participants confirmed their understanding and agreed to these basic requirements for the online experiment.

Participants were pseudo-randomly assigned to either the high-stress (experimental) condition ($n$ = 82) or the low-stress (control) condition ($n$ = 55). The term "pseudo-randomly" reflects the use of a computer-based randomization algorithm, which generates values based on a pseudo-random number generator. While this method approximates randomness, it is not entirely random in the strict mathematical sense due to the inherent limitations of algorithmic randomization. In addition, based on the cut-off score of the Beck Depression Inventory-II (BDI-II), participants were classified into the depressive group (BDI-II score > 13) or the non-depressive group (BDI-II score < 14) within each condition. The final distribution was as follows: in the high-stress condition, 48 participants were classified as depressive and 34 as non-depressive, and in the low-stress condition, 21 were classified as depressive and 34 as non-depressive. Due to technical issues, the allocation of participants to the high-stress and control groups did not converge to an equal distribution, resulting in a slightly larger number of participants in the experimental group.

## Measures

### *Beck depression inventory-II*

BDI-II is a 21-item self-report scale that evaluates the severity of depressive symptoms (*Beck, Steer & Brown, 1996*). Each item is rated on a 4-point scale ranging from 0 to 3. Participants completed the Japanese version of BDI-II, which has well validation and reliability (*Kojima et al., 2002*). The internal consistency in this study was $\alpha$ = 0.929.

The cut-off score of the BDI-II was set as the total score ≥ 14. The cut-off score of 14 was chosen as it aligns with the thresholds established in previous research (*Karimpour-Vazifehkhorani et al., 2020*; *Mura et al., 2023*) and recommendations for identifying mild depressive symptoms in non-clinical populations (*Kojima et al., 2002*;

*Beck, Steer & Brown, 1996*). This cut-off ensures sensitivity in capturing participants with depressive tendencies while acknowledging that scores ≥ 14 include mild, moderate, and severe depressive symptoms. This approach is consistent with the goals of the study, which aimed to investigate cognitive control mechanisms in individuals across a broad spectrum of depressive symptoms.

### AX-version continuous performance task

The AX-CPT is a cognitive task designed to measure the degree of reactive and proactive control processes (*Braver, Barch & Cohen, 1999*). Cognitive control is assessed by determining whether participants can appropriately respond to a probe stimulus while retaining information from a previously presented cue stimulus (*Braver, 2012*). Cue and probe stimuli are presented as letters from the alphabet. Participants are instructed to press the "target" key ("j") when the probe letter "X" is preceded by the correct cue letter "A" (AX trial). In all other cases—"A" followed by a non-"X" (AY trial), a non-"A" followed by "X" (BX trial), or both non-"A" and non-"X" (BY trial)—participants are required to press the "non-target" key ("f").

Each trial consists of the following sequence (see Fig. 1): a fixation cross (500 ms), the presentation of the cue stimulus (500 ms), a blank screen with fixation (4,000 ms), and the presentation of the probe stimulus (until response or a 2,000 ms timeout). The fixation, cue, and probe letters are displayed in approximately 70-point Arial font. The task includes 100 trials in total, with 70 AX trials and 10 trials each for the AY, BX, and BY conditions.

### Anagram task

The Anagram task was used to induce an experimental stress state. Participants were asked to rearrange five Japanese hiragana characters into meaningful words. The stimuli were sourced from a database of 5-letter Hiragana Anagrams (*Ichimura, Ueda & Kusumi, 2017*). The difficulty level of the anagrams was manipulated to create high- and low-stress conditions. Based on the difficulty indices—correct rate, solving time, and subjective difficulty—from the database (*Ichimura, Ueda & Kusumi, 2017*), stimuli were selected for each stress condition. For the high-stress group, the average correct rate was 73%, solving time was 49.12 s, and subjective difficulty was 4.61 (on a scale from 1 to 7). In contrast, the low-stress group had an average correct rate of 98%, solving time of 10.65 s, and subjective difficulty of 1.84.

Both the high- and low-stress conditions consisted of 10 anagrams. Participants were given 30 s to solve each anagram by typing the correct letters. If the 30 s time limit was exceeded, the next anagram automatically began.

### Visual analogue scale

To assess the stress induced by the anagram task, three visual analogue scales (VAS), each ranging from 1 to 100, were used. These scales measured: Stress (from "no stress" to "severe stress"), Fatigue (from "not tired" to "very tired"), and Mood (from "sad" to "happy"). Participants rated their current state on each scale according to how they felt at the time.

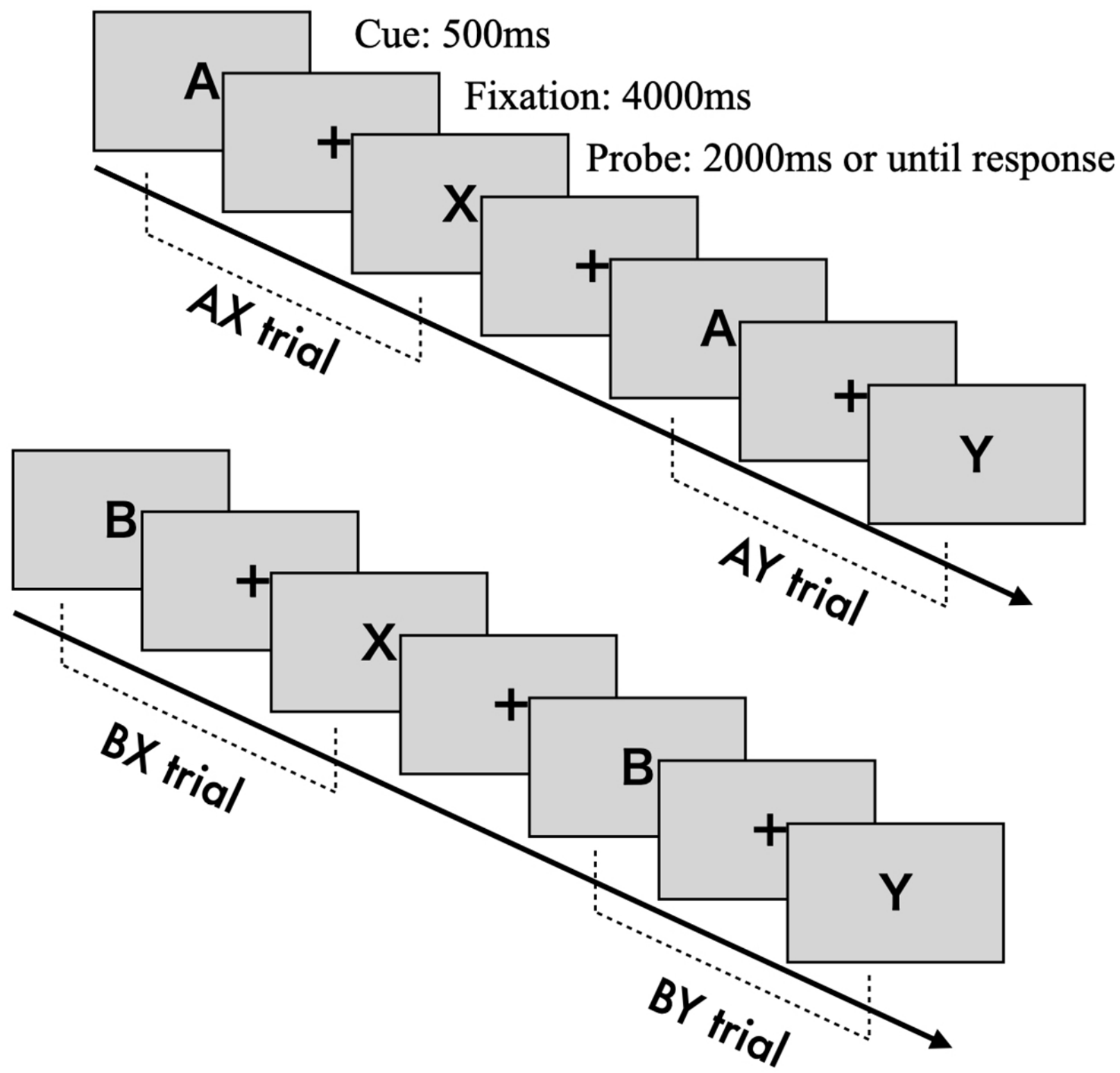

**Figure 1** **The construction of AX-CPT.**

## Procedures

Figure 2 outlines the study procedures. First, participants provided informed consent and completed the Beck Depression Inventory (BDI). Then, the AX-CPT and VAS assessments were conducted as baseline measurements. Participants pseudo-randomly assigned to the high-stress group completed difficult anagrams, while those in the control group solved easier anagrams. The VAS was repeated to assess stress manipulation. Finally, the AX-CPT was administered again. The experiment concluded with a debriefing for both groups, and participants in the high-stress group were shown a simple relaxation method to alleviate their stress. The procedures were conducted online. All study procedures were approved by the ethics committee of the Aichi University of Education (approval no. AUE20240203HUM).

## Data analyses

The error rates and average reaction times in the AX-CPT were recorded for each trial type: AX, AY, BX, and BY. These data were used to compute three indices that reflect cognitive control activity: the Proactive Behavior Index (PBI), *d'*-context, and A-cue bias. The PBI, calculated as (AY − BX)/(AY + BX) using error rates and reaction times, reflects the balance between proactive and reactive control (*Braver et al., 2009*). Higher cue interference, leading to more errors and delayed responses in AY trials, indicates a dominance of proactive control. Conversely, higher probe interference in BX trials suggests a dominance of reactive control. Therefore, a positive PBI value indicates a greater reliance on proactive control, while a negative value suggests a shift toward reactive control.

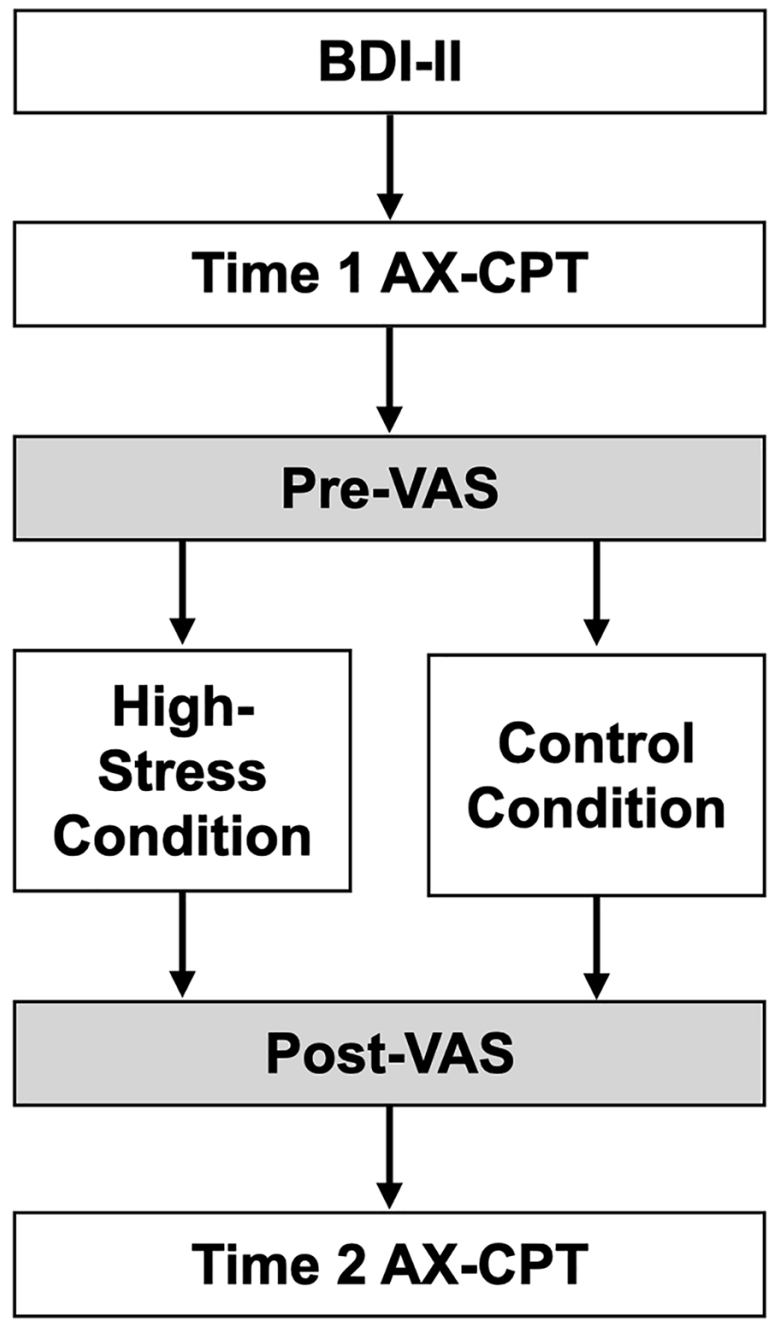

**Figure 2 The experimental procedure in this study.** Note. BDI, Beck Depression Inventory; AX-CPT, AX version Continuous Performance Task; VAS, Visual Analogue Scale.

The *d*'-context and A-cue bias indices are based on signal detection theory. *d*'-context was calculated as Z(AX hit rate) – Z(BX error rate), with Z representing the z-transformation. This index reflects the ability to discriminate between the correct cue ("A") and noise ("B"). A higher *d*'-context value indicates better retention of contextual information, while a lower value reflects poorer retention. A-cue bias was calculated as 1/2 * (Z(AX hit rate) + Z(AY error rate)). This index measures participants' ability to form correct responses based on cue information ("A"), with higher values indicating more accurate cue-based responses regardless of the probe stimuli.

Exclusion criteria, based on the performance in the AX-CPT, were adopted following the guidelines from Gonthier et al. (2016) and Snijder et al. (2024). Fourteen participants were excluded from the analysis due to extremely poor performance: five had an error rate of 50% or more in AX trials, four had an error rate of 90% or more in AY, BX, and BY trials, and five had an average error rate of 40% or more in AY, BX, and BY trials.

Mixed ANOVAs were conducted for each variable. To improve normality, log-transformed reaction times and z-transformed error rates were used in the ANOVAs for AX-CPT performance. Assumptions for normality, homogeneity of variance, and sphericity were tested for all variables prior to the main analyses. Normality was assessed using the Shapiro-Wilk test, with most conditions satisfying this assumption. For conditions where normality was not satisfied, I applied Fisher's Z-transformation, as described earlier. Regarding homogeneity of variance, the z-transformation of error rates inherently stabilizes variances across groups, and Levene's test confirmed no significant violations ($p > 0.05$ for all comparisons). Sphericity was assessed using Mauchly's test, and

Greenhouse-Geisser corrections were applied when violations were detected. The Tukey correction was applied in *post hoc* analyses to control for Type I error rates in multiple comparisons.

# RESULTS

## Manipulation check

The descriptive statistics were summarized in Table 1. Firstly, to confirm the experimental manipulation, repeated ANOVAs were conducted for each VAS (mood, stress, and fatigue) with time (pre/post) and condition (high stress/control condition; Fig. 3). For VAS of stress, the analysis revealed the significant main effects of time and condition ($F$ (1, 124) > 10.554, $p \leq 0.001$, $\eta p^2 > 0.078$) and their interaction ($F$ (1, 124) = 32.201, $p < 0.001$, $\eta p^2 = 0.206$). The *post hoc* analysis showed that the VAS of stress in high-stress condition on post-timepoint was greater than the VAS in high-stress condition on pre-timepoint ($p < 0.001$, Cohen's $d = 0.78$), and the VAS in the control condition on post-timepoint ($p < 0.001$, Cohen's $d = 1.15$). For VAS for fatigue, the significant main effect of condition and interaction of time and condition were also observed ($F$s (1, 124) > 12.766, $p$s < 0.001, $\eta p^2 > 0.093$). The *post hoc* analysis revealed that the VAS of fatigue in high stress condition on post-timepoint was greater than on pre-timepoint ($p < 0.001$, Cohen's $d = 0.47$), and the VAS in control condition on post-timepoint ($p < 0.001$, Cohen's $d = 1.04$). However, the ANOVA for VAS of mood revealed a non-significant interaction of time and condition ($F$ (1, 124) = 0.273, $p = 0.776$, $\eta p^2 = 0.000$), suggesting that the experimental manipulation of stress state in this study did not affect the VAS of mood. Through all ANOVAs for VAS of stress, fatigue, and mood, there were no significant main effects of group ($F$s (1, 124) > 3.652, $p$s < 0.058), showing the irrelevance of depressive symptoms for the induction of stress states. Summing up, the results showed that the participants in high stress condition felt stress and mental fatigue compared to before they administered the difficult anagram task, and to participants in control condition, suggesting the success of experimental manipulation.

## Performances of AX-CPT

Secondly, to identify the effect of stress on AX-CPT performance, repeated measures ANOVAs were conducted for error rates and reaction times, with the following independent variables: trial type (AX, AY, BX, BY; within-subject), time (Time 1/Time 2; within-subject), condition (high stress/control; between-subject), and group (depressive/non-depressive; between-subject). The analysis of reaction times revealed a significant main effect of trial type ($F$ (3, 124) = 234.913, $p < 0.001$, $\eta p^2 = 0.655$) and time ($F$ (1, 124) = 29.908, $p < 0.001$, $\eta p^2 = 0.194$), as well as an interaction of time and condition ($F$ (3, 124) = 4.248, $p = 0.041$, $\eta p^2 = 0.033$). However, none of the other main effects or interactions were significant ($F$s > 3.767, $p$s < 0.055). The *post hoc* test revealed that the reaction times for the AY trial were significantly greater than for the AX, BX, and BY trials ($p$s < 0.001, Cohen's $d$s > 1.366). The analysis also showed that the reaction times at Time 1 for participants in the non-depressive group and control condition were significantly greater than at Time 2 ($p < 0.001$, Cohen's $d = 0.406$). Similarly, the reaction times at Time

**Table 1** The descriptive statistics.

| | Control condition | | High stress condition | |
|---|---|---|---|---|
| | Non-depressive group ($n$ = 34) | Depressive group ($n$ = 20) | Non-depressive group ($n$ = 45) | Depressive group ($n$ = 23) |
| Age | 40.77 (9.60) | 43.15 (8.34) | 40.84 (7.87) | 38.21 (7.29) |
| Sex (Female rate) | 47.06% | 40.00% | 33.33% | 27.59% |
| BDI score | 5.56 (4.60) | 21.55 (4.90) | 5.20 (4.58) | 21.45 (6.64) |
| Time 1 | | | | |
| RT (ms) | | | | |
| AX | 480.39 (72.6) | 485.19 (78.50) | 490.388 (76.66) | 459.54 (61.29) |
| AY | 608.94 (74.00) | 611.29 (79.05) | 589.53 (72.25) | 583.72 (90.41) |
| BX | 481.791 (108.98) | 489.07 (107.20) | 475.74 (112.25) | 431.39 (79.53) |
| BY | 483.05 (87.62) | 489.59 (89.90) | 468.94 (95.81) | 453.55 (89.35) |
| Error rate | | | | |
| AX | 0.058 (0.052) | 0.069 (0.068) | 0.078 (0.077) | 0.105 (0.106) |
| AY | 0.162 (0.148) | 0.145 (0.094) | 0.144 (0.124) | 0.166 (0.161) |
| BX | 0.153 (0.176) | 0.185 (0.239) | 0.16 (0.185) | 0.19 (0.172) |
| BY | 0.035 (0.069) | 0.04 (0.099) | 0.02 (0.059) | 0.021 (0.056) |
| Time 2 | | | | |
| RT (ms) | | | | |
| AX | 440.69 (59.29) | 469.19 (80.39) | 461.21 (76.34) | 445.02 (52.21) |
| AY | 567.85 (103.64) | 596.20 (97.94) | 586.35 (100.20) | 580.52 (88.70) |
| BX | 420.10 (93.91) | 450.12 (123.77) | 443.93 (118.69) | 435.28 (100.52) |
| BY | 430.72 (75.89) | 459.25 (105.91) | 437.46 (111.70) | 429.44 (82.81) |
| Error rate | | | | |
| AX | 0.035 (0.054) | 0.075 (0.086) | 0.055 (0.076) | 0.046 (0.053) |
| AY | 0.185 (0.199) | 0.15 (0.095) | 0.211 (0.2) | 0.183 (0.207) |
| BX | 0.106 (0.135) | 0.2 (0.232) | 0.124 (0.177) | 0.141 (0.127) |
| BY | 0.015 (0.044) | 0.045 (0.076) | 0.016 (0.037) | 0.034 (0.081) |

**Note:**
BDI, Beck Depression Inventory; RT, Reaction Time.

1 for participants in the depressive group and high-stress condition were significantly greater than at Time 2 ($p$ = 0.037, Cohen's $d$ = 0.184). These results suggest that improvements in reaction time due to habituation from repeated AX-CPT trials occurred regardless of condition and group.

For the error rates, the same analysis revealed a significant main effect of trial type ($F$ (3, 124) = 55.956, $p < 0.001$, $\eta p^2 = 0.311$) and an interaction between trial type and time ($F$ (3, 124) = 4.075, $p$ = 0.007, $\eta p^2 = 0.032$). In contrast, none of the other main effects or interactions were significant ($F$s < 2.467, $p$s > 0.119). The *post hoc* analysis showed that the error rates for the AY and BX trials were significantly greater than those for the AX and BY trials ($p$s < 0.001, Cohen's $d$s > 0.706). Similar results were obtained from the *post hoc* analysis for the interaction between trial type and time ($p$s < 0.001, Cohen's $d$s > 0.586), suggesting consistently higher error rates for both AY and BX trials. These results indicate

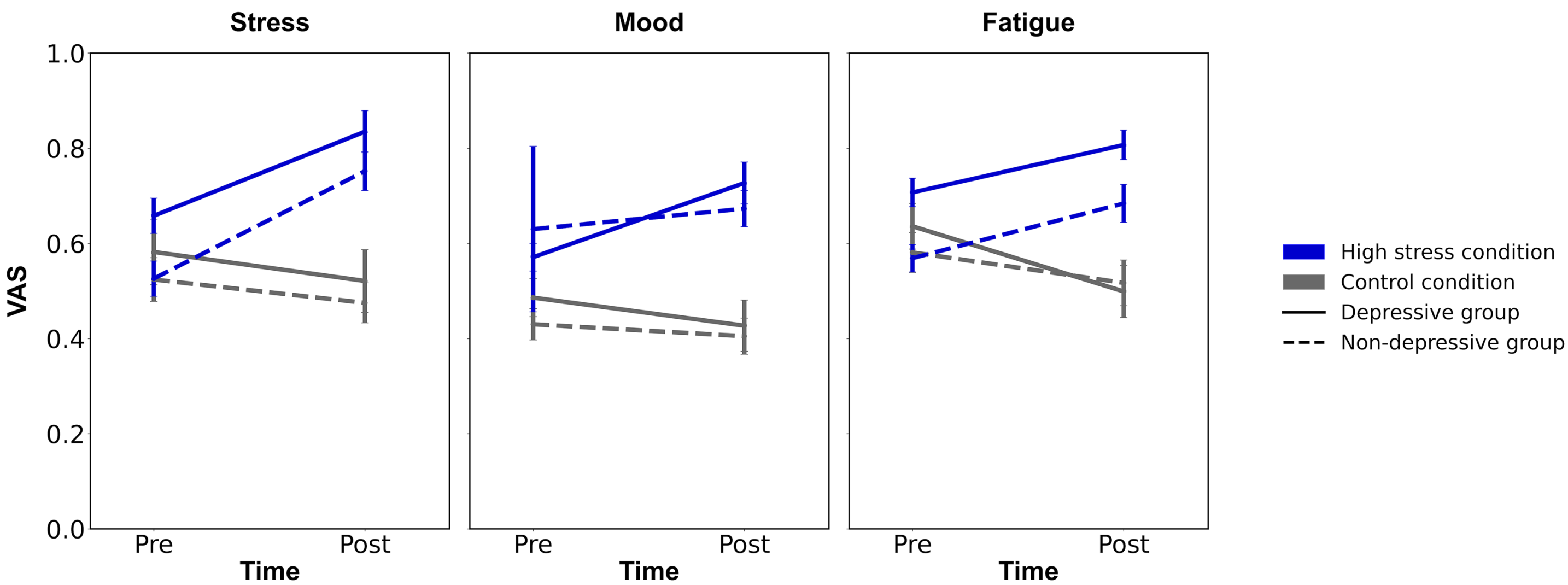

**Figure 3 The results of experimental manipulation.** Note. VAS, Visual Analogue Scale; Error bars represent ±1 Standard Error.

a specificity of the AY and BX trials in the AX-CPT, supporting the validity of computed indices such as PBI, which is discussed below.

## Analyses of cognitive control metrics

Lastly, repeated ANOVAs were conducted on each index: PBI, *d*'-context, and A-cue bias (Table 2; Fig. 4). The analyses included the factors of time (Time 1/Time 2), condition (high stress/control), and group (depressive/non-depressive). The analysis for PBI showed that none of the main effects or interactions reached statistical significance ($F$s (1, 124) < 0.722, $p > 0.08$, $\eta p^2 < 0.024$). The result suggested that there were no significant changes in the balance between reactive and proactive control before and after experiencing stress, regardless of the presence or absence of depressive symptoms.

The analysis for *d*'-context revealed a significant main effect of time ($F$ (1, 124) = 6.940, $p < 0.001$, $\eta p^2 = 0.012$) and a three-way interaction of time, condition, and group ($F$ (1, 124) = 4.739, $p = 0.03$, $\eta p^2 = 0.004$). The *post hoc* analysis showed that the score in the high-stress condition for the depressive group at Time 2 was lower than that at Time 1 ($p = 0.026$, Cohen's $d = 0.531$) and also lower than the score for the non-depressive group in the control condition at Time 2 ($p = 0.020$, Cohen's $d = 0.852$). The results suggested that individuals in the depressive group under high stress experienced a greater decline in *d*'-context from Time 1 to Time 2, with their scores being lower at Time 2 compared to those in the non-depressive group in the control condition. This indicates that the combination of high stress and depressive symptoms may have a more pronounced negative impact on cognitive control over time.

The analysis for A-cue bias showed a significant main effect of time ($F$ (1, 124) = 13.296, $p < 0.001$, $\eta p^2 = 0.097$). However, none of the other main effects or interactions reached statistical significance ($F$s (1, 124) < 1.867, $p > 0.17$, $\eta p^2 < 0.015$). The *post hoc* analysis revealed that the score in the high-stress condition at Time 2 was significantly lower than

**Table 2** The statistics of each index of cognitive control.

| | Control condition | | High stress condition | |
|---|---|---|---|---|
| | Non-depressive group ($n$ = 34) | Depressive group ($n$ = 18) | Non-depressive group ($n$ = 20) | Depressive group ($n$ = 18) |
| Time 1 | | | | |
| PBI for RT | 0.064 (0.045) | 0.068 (0.036) | 0.059 (0.048) | 0.077 (0.036) |
| PBI for ER | 0.066 (0.514) | 0.037 (0.512) | 0.051 (0.462) | −0.057 (0.501) |
| d'-context | 2.889 (0.921) | 2.736 (1.164) | 2.798 (1.163) | 2.487 (1.077) |
| A-cue bias | 0.313 (0.353) | 0.268 (0.283) | 0.252 (0.341) | 0.191 (0.516) |
| Time 2 | | | | |
| PBI for RT | 0.076 (0.059) | 0.08 (0.052) | 0.069 (0.044) | 0.064 (0.061) |
| PBI for ER | 0.171 (0.548) | 0.036 (0.595) | 0.225 (0.54) | 0.054 (0.545) |
| d'-context | 3.383 (0.865) | 2.742 (1.359) | 3.114 (1.043) | 3.046 (0.86) |
| A-cue bias | 0.506 (0.406) | 0.305 (0.382) | 0.443 (0.421) | 0.415 (0.511) |

**Note:**
PBI, Proactive Behavior Index; RT, Reaction Time; ER, Error rate.

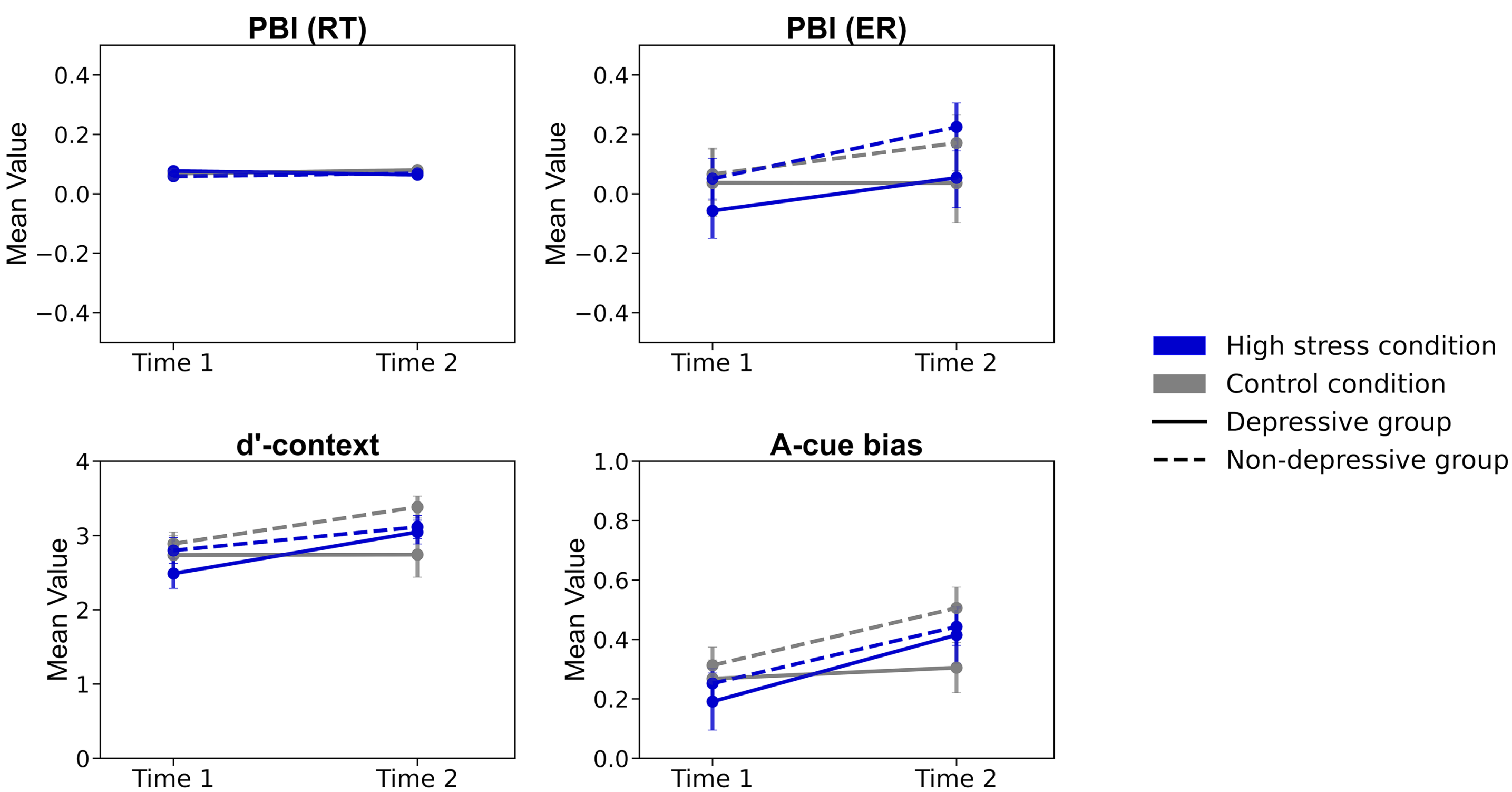

**Figure 4** **The results of each index of cognitive control.** Note. PBI (RT), Proactive Behavior Index for Reaction Time; PBI (ER), Proactive Behavior Index for Error rate; Error bars represent ±1 Standard Error.

that at Time 1 ($p$ = 0.002, Cohen's $d$ = 0.508). The results suggested that A-cue bias decreased after experiencing high stress, regardless of the presence or absence of depressive symptoms.

## DISCUSSION

The aim of this study was to investigate the impact of experimental stress on cognitive control within the context of depression psychopathology. To achieve this, participants were divided into high-stress and control conditions and further classified into depressive and non-depressive groups based on the BDI cutoff score. The AX-CPT task was administered both before and after exposure to the experimental stressor in the high-stress condition, while the control condition involved exposure to a non-stressful or low-stress environment. Subsequently, the effects of stress on depressive symptoms and cognitive control were examined by comparing AX-CPT performance and cognitive control measures after stress in the depressed group with their performance before stress, as well as with the control group. The analyses revealed that the balance between reactive and proactive control, as conceptualized in the Dual Mechanisms of Control (DMC) framework, did not change after exposure to higher experimental stressors. However, the function of cognitive control related to retaining contextual information decreased after exposure to experimental stress in the depressive group. This indicates that stress selectively impacts aspects of cognitive control associated with the psychopathology of depression.

The statistical analysis revealed that the PBI in the high-stress condition for the depressive group did not differ from any other conditions or groups. This suggests that the balance between reactive and proactive control was not affected by stress in relation to depressive symptoms consistent with findings from previous studies (*Birk et al., 2018*; *Husa, Buchanan & Kirchhoff, 2022*). *Birk et al. (2018)* examined the effect of proactive control training on anxiety-related characteristics. Their results showed that proactive control training did not improve PBI scores; however, state anxiety increased less during social stress tasks compared to the control condition. Similarly, *Husa, Buchanan & Kirchhoff (2022)* found no differences in PBI scores, calculated using both reaction times and error rates, before and after exposure to the experimental stressor, despite evidence of physiological stress changes. Taken together, these findings suggest that changes in the balance of cognitive control, as reflected in the PBI, may depend on more stable factors and conditions rather than those that induce transient fluctuations. Indeed, *Cooper et al. (2017)* reported that individuals diagnosed with schizophrenia exhibited stable performance in the AX-CPT compared to healthy controls, indicating a consistent influence of psychiatric-related factors on reactive and proactive control. Therefore, to more precisely identify the effects of stress on the balance between reactive and proactive control, it would be necessary to focus on relatively long-term time points.

The analysis for PBI did not reveal a main effect of group (depressive/non-depressive), suggesting that depressive symptoms is not related to the balance between reactive and proactive control. While well-documented links between cognitive control and depression exist (*e.g.*, *Koster et al., 2017*; *Paulus, 2015*; *Villalobos, Pacios & Vázquez, 2021*), research specifically examining the relationship between PBI and depressive symptoms remains limited. Therefore, the lack of an association between PBI and depressive symptoms in this study is noteworthy. A study by *Masuyama & Mochizuki (2020)* similarly found no

relationship between PBI and depressive symptoms, aligning with our findings. However, other studies have demonstrated a link between depression-related characteristics and cognitive control within the DMC framework (*Masuyama et al., 2018*; *Vanderhasselt et al., 2014*). Given the lack of association between PBI and depressive symptoms in this study, there are two possible explanations. First, the validity of PBI as an index for the balance between reactive and proactive control may be in question. As *Snijder et al. (2024)* suggested, PBI—calculated solely based on performance in AY and BX trials—may require an alternative or supplementary indicator. Second, impaired reactive or proactive control processes may be specific to negative-related cognitive processes, consistent with the distinction between "hot" and "cold" cognition (*Roiser & Sahakian, 2013*). Indeed, *Masuyama et al. (2018)* reported that cognitive control impairments in depressive individuals were only evident when negative-valenced distractors were present. Thus, the relationship between depression and cognitive control within the DMC framework still requires further exploration, and additional research is necessary to advance our understanding.

However, partial effects of both stress and depressive symptoms on cognitive control were suggested by the results of the *d*'-context index in this study. In contrast, the results for A-cue bias did not reach statistical significance. Considering the nature of these indices—*d*'-context being related to the retention of cue information, while A-cue bias pertains to probe-response processes—the findings can be interpreted as the interaction of depressive symptoms and stress primarily affecting cue-related information processing. As schematically depicted in *Braver (2012)* and *Li et al. (2018)*, from cue presentation to probe presentation, individuals need to activate proactive control, which is involved in goal activation and maintenance. Then, from probe presentation to response, they must activate reactive control, which is involved in conflict resolution and response inhibition. Notably, the switch from proactive to reactive control at the time of probe presentation is considered crucial for appropriate cognition and behavior (*Braver, 2016*). Therefore, the results of this study, where the interaction between depressive symptoms and stress was only associated with cue-related processes but not with probe-related processes, likely reflect impaired proactive control. Specifically, the findings suggest that depressive individuals exhibit impaired proactive control after exposure to stress. Indeed, several studies have reported that depressive individuals exhibit deficits in maintaining contextual information (*Masuyama et al., 2018*; *Msetfi et al., 2009*). Given the possibility that reactive and proactive control processes are not entirely independent, the results of this study suggest that depressive symptoms and stress are associated with impaired proactive control.

## STUDY LIMITATIONS

In the end, this study had several limitations. First, because this study was conducted on a general population, caution is needed when extending the interpretation of the results to clinically depressed patients. Some previous studies have demonstrated the influence of psychiatric conditions on cognitive control among patients (*Jia et al., 2023*; *Kwashie et al., 2023*; *Lesh et al., 2013*). These studies compared patients with healthy controls and

therefore may have accurately detected differences in cognitive control. Future research on the cognitive control mechanisms of depression should be conducted on clinically diagnosed individuals.

Second, this study was experimental and did not examine the effects of medium-term or long-term stress. As previously mentioned, the actual effects of prolonged stress on cognitive control should be investigated in more detail in future studies.

Third, as an online study, this research may involve inherent biases related to self-selection and socioeconomic status, which could influence the generalizability of the findings. For example, factors such as educational background, occupation, and access to resources (*e.g.*, stable Internet connections or familiarity with digital platforms) were not collected or controlled in this study, and these demographic variables could potentially influence cognitive control performance. Future studies should aim to collect and control for such demographic data to provide a more nuanced understanding of how socioeconomic factors interact with cognitive control mechanisms in depression.

Lastly, the study relied on self-reported measures for assessing stress and fatigue levels, which may introduce subjective biases. While these measures are widely used in psychological research, they may not fully capture physiological stress responses, such as changes in cortisol levels or heart rate variability. Future studies should consider incorporating objective biomarkers to provide a more comprehensive understanding of stress impacts.

## CONCLUSIONS

This study aimed to examine the effects of experimental stress on cognitive control within the context of depression. By dividing participants into high-stress and control conditions and further classifying them into depressive and non-depressive groups, I sought to understand how stress interacts with depressive symptoms to affect cognitive processes. Using the AX-CPT task, I found that while the balance between reactive and proactive control remained unaffected by stress at the group level, individuals in the depressive group exhibited a decline in their ability to retain contextual information following stress exposure.

These findings indicate that stress may selectively impair proactive control mechanisms in individuals with depressive tendencies. However, given the study's limitations, such as the use of a non-clinical sample and the reliance on short-term stress manipulations, these conclusions should be interpreted with caution. Future research should investigate these effects in clinically diagnosed populations to better generalize the findings to depression psychopathology. Additionally, longitudinal studies examining the impact of prolonged stress on cognitive control are needed to clarify the temporal dynamics of stress-related impairments. Incorporating diverse cognitive tasks and objective stress markers, such as cortisol or neural measures, could also provide more robust evidence regarding the mechanisms linking stress and cognitive control deficits in depression.

## ACKNOWLEDGEMENTS

This study utilized a generative AI/large language model provided by OpenAI to assist with text proofreading and editing support. The author takes full responsibility for the interpretation and content presented in this work.

### Funding

This work was supported by the Japan Society for the Promotion of Science (JSPS) KAKENHI Grant Number (No. 22K13839). The funders had no role in study design, data collection and analysis, decision to publish, or preparation of the manuscript.

### Grant Disclosures

The following grant information was disclosed by the authors:
Japan Society for the Promotion of Science(JSPS) KAKENHI: 22K13839.

### Competing Interests

The author declares that they have no competing interests.

### Author Contributions

- Akihiro Masuyama conceived and designed the experiments, performed the experiments, analyzed the data, prepared figures and/or tables, authored or reviewed drafts of the article, and approved the final draft.

### Human Ethics

The following information was supplied relating to ethical approvals (*i.e.*, approving body and any reference numbers):

The Ethical Committee of Aichi University of Education granted Ethical approval to carry out the study (Ref: AUE20240203HUM).

### Data Availability

The raw data is available in the Supplemental File.

### Supplemental Information

Supplemental information for this article can be found online at http://dx.doi.org/10.7717/peerj.18821#supplemental-information.

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
