# Peer review of "The impact of induced stress on reactive and proactive control in depression"

_PeerJ, doi:10.7717/peerj.18821_

## Round 0.1 · original submission · Major Revisions

Please address all the comments from both reviewers in detail

·

Basic reporting

1-)background seems good.
2-)you can improve the following sentence and you can avoid having absolute results:
This suggests that depressive individuals are particularly vulnerable to stress-induced impairments
in proactive control.
3-)explain the models.
it encompasses a range of functions and has been conceptualized through
68 various models
4-)you should add more references.
Several studies have demonstrated that depressive symptoms are closely linked with
77 deficits in cognitive control (Joormann & Tanovic, 2015).
5-)you can add links as a reference.
6-)you can use more updated references.
7-)you can add more limitations related to the methods .

Experimental design

acceptable

Validity of the findings

acceptable

Additional comments

-

Reviewer 2 ·

Basic reporting

no comment

Experimental design

The current article explores how stress and depression interact to affect cognitive control. It used the AX-Continuous Performance Task (AX-CPT) as a measurement tool. Participants were divided into high-stress and control groups, as well as depressive and non-depressive groups, based on their Beck Depression Inventory II (BDI-II) scores. The study conclude that stress selectively impacts cognitive control, mainly in depressed individuals, affecting their ability to keep contextual information. The study is an original research study within the scope of the journal. It needs several revisions.
1. The hypothesis and aim of this study was clearly articulated in the introduction section. However, authors should justify why they prefer anagram task a source of stress induction over other paradigms or other tasks.
2. The study is an online research which inherently brings forward some biases related to self-selection and socioeconomic status. The demographic variables that could influence cognitive control (e.g., educational background, occupation) has not been mentioned and not controlled in analysis. These issues should be considered as limitations if no such data present.
3. In some parts of article, the terms depressive has been used. As it is a clinical diagnosis, the term higher (or lower) depression scores instead of depression terms should be preferred.
4. Why did the authors choose to use 13 as cut-off score of the BDI-II to classify participants as depressive or non-depressive? What is their rationale or reference? As scores higher than 13 comprises of all types of depression classification ( mild, moderate, or severe)I, it should be explained in detail.
5. As the study was an online study, it has several more limitations which brings some concerns over the level of control over the experimental environment. How did the authors arrange participants' conditions (e.g., distractions, screen settings, internet speed) that could effect outcome of study? Provide the precautions you have taken to control or reduce such problems.
6. Define what you meant by pseudo randomly distribution in methods section .
7. Clarify how you addressed the risk of overfitting or multiple comparisons in mixed ANOVA testing. Also provide what you have done to test assumptions such as normality, sphericity, or homogeneity of variance
8. Authors should make cautious comments based on results. The lack of significant differences between groups (depressive/non-depressive) and conditions (high stress/control) in many analyses is reported as though it were a meaningful finding. However without reporting effect size or statistical power, it would be hard to make such claims. Please revise section accordingly.
9. More limitations like non controlling of demographic variables, nonrepresentative sample, use of self report scales should be considered and discussed .
10. The conclusions should be rewritten as it is disconnected from the data presented. The authors should be more cautious in making definitive statements based on limitations. More concrete suggestions for future study should be added.

Validity of the findings

see above

Additional comments

no comments

---

## Round 0.2 · Minor Revisions

First, congratulations on a clearly-written and interesting article, and I concur that you have adequately addressed concerns raised by the reviewers at the previous rounds.

I have two more items which need to be addressed before the article is ready for acceptance.

1. In many of the statistical results, ANOVAs are reported where the degrees of freedom seem implausible: e.g. line 272 reports F(124,1) for a study with 2 factors and 123 participants (based on the dataset included with the submission). Typically, when the first degree of freedom (between-subjects df) is so much larger than the second (within-subjects df), this suggests the ANOVA may have been mis-specified – in which case the F-value, p-value, and general conclusions would all be incorrect. This occurs in many (but not all!) the ANOVAs reported in the article. It is possible that (in some cases) a “typo” resulted in swapping the conventional order of the df … although I am still not sure how the within-subjects df can exceed number of participants. Please doublecheck the ANOVAs and either correct/explain the unusual df, or (if needed) re-run the analyses to confirm findings and results.

2. PeerJ typically requires a measure of variability (e.g. error bars) on figures that present measures of central tendency such as mean/median. Please add error bars (typically serr or 95% CI) for each group/condition in Figures 3 and 4, or alternately show the raw data (e.g. overlay strip plot).

I hope that these will be straightforward to address (hence my editorial determination as "Minor Revisions"), but it is possible that they (especially #1) will meaningfully change the results -- in which case this could constitute a major revision and I would likely send the article out for re-review.

·

Basic reporting

I accept.

Experimental design

acceptable

Validity of the findings

acceptable

Additional comments

acceptable

Reviewer 2 ·

Basic reporting

Authors responded all of my comments adequately.

Experimental design

No comment

Validity of the findings

No comment

---

## Round 0.3 · accepted · Accept

I'm sincerely glad that the statistics issue turned out to be easily corrected, and the results if anything look even stronger with error bars on the figures. I think this paper is now ready for publication, and will make a nice contribution to the field.